# Genome-Wide Identification and Functions against Tomato Spotted Wilt Tospovirus of PR-10 in *Solanum lycopersicum*

**DOI:** 10.3390/ijms23031502

**Published:** 2022-01-28

**Authors:** Md. Monirul Islam, Shiming Qi, Shijie Zhang, Bakht Amin, Vivek Yadav, Ahmed H. El-Sappah, Fei Zhang, Yan Liang

**Affiliations:** 1College of Horticulture, Northwest A&F University, Xianyang 712100, China; monirul@nwafu.edu.cn (M.M.I.); qishiming2008@nwafu.edu.cn (S.Q.); 13289316916zsj@nwafu.edu.cn (S.Z.); bakhtamin96@nwafu.edu.cn (B.A.); vivekyadav@nwafu.edu.cn (V.Y.); ahmed_elsappah2006@yahoo.com (A.H.E.-S.); feizhang@nwsuaf.edu.cn (F.Z.); 2State Agriculture Ministry Laboratory of Northwest Horticultural Plant Germplasm Resources & Genetic Improvement, Northwest A&F University, Xianyang 712100, China; 3Genetics Department, Faculty of Agriculture, Zagazig University, Zagazig 44511, Egypt

**Keywords:** PR-10 protein, resistance, *Sw-5b*, tomato, TSWV

## Abstract

Tomato spotted wilt virus impacts negatively on a wide range of economically important plants, especially tomatoes. When plants facing any pathogen attack or infection, increase the transcription level of plant genes that are produced pathogenesis-related (PR) proteins. The aim of this study is a genome-wide identification of PR-10 superfamily and comparative analysis of *PR-10* and *Sw-5b* gene functions against tomato responses to biotic stress (TSWV) to systemic resistance in tomato. Forty-five candidate genes were identified, with a length of 64–210 amino acid residues and a molecular weight of 7.6–24.4 kDa. The *PR-10* gene was found on ten of the twelve chromosomes, and it was determined through a genetic ontology that they were involved in six biological processes and molecular activities, and nine cellular components. Analysis of the transcription level of PR-10 family members showed that the *PR-10* gene (Solyc09g090980) has high expression levels in some parts of the tomato plant. *PR-10* and *Sw-5b* gene transcription and activity in tomato leaves were strongly induced by TSWV infection, whereas H8 plants having the highest significantly upregulated expression of *PR-10* and *Sw-5b* gene after the inoculation of TSWV, and TSWV inoculated in M82 plants showed significantly upregulated expression of *PR-10* gene comparatively lower than H8 plants. There was no significant expression of *Sw-5b* gene of TSWV inoculated in M82 plants and then showed highly significant correlations between *PR-10* and *Sw-5b* genes at different time points in H8 plants showed significant correlations compared to M82 plants after the inoculation of TSWV; a heat map showed that these two genes may also participate in regulating the defense response after the inoculation of TSWV in tomato.

## 1. Introduction

When pathogens infect a plant, pathogenesis-related proteins are produced. They are now categorized into 17 groups based on structural and functional features, and they play a key role in plant growth and development under a variety of stressors [1]. Through their enzymatic activity, PR proteins can directly impact pathogen integrity and/or create signal molecules, which act as a trigger to activate other plant defense processes [2]. PR-10 is a naturally occurring acidic protein with a molecular weight of 16–19 kDa and three-dimensional β-sheet topologies covered by a compact, bipartite framework held together by hydrophobic interactions. PR-10 proteins have antimicrobial action and are involved in a variety of biological processes, including ribonuclease and other secondary metabolism enzymatic functions, along with plant defense against biotic and abiotic stressors [3]. The majority of PR-10 proteins are made up of two domains: Bet_ v _1 and P-loop. Bet_ v _1 is a conserved domain of the PR-10 protein family that serves as a defense against pathogen infection. The RNase activity of these proteins contains a P-loop motif, such as GxGGxGxxK, which serves as a nucleotide-binding site [4].

Tomatoes (*Solanum lycopersicum*) are the most widely grown commercial vegetable crop in the world. The tomato plant, on the other hand, is susceptible to a variety of diseases. More than 136 viral species that are detrimental to tomatoes have been discovered so far [5]. Tomato spotted wilt virus (TSWV) is one of the most dangerous. TSWV cause significant reductions in tomato output and market value, as well as mortality in certain cases [6]. Usually, the entire plant is dwarfed, with necrotic streaks and dark brown flecks on the leaves, stems, and fruits. The first symptoms of seedlings inhibited the growth points and young leaves turn into copper-colored rolls, and subsequently form many small dark brown flecks, the leaves’ veins will be purple. Furthermore, the virus causes growth points leaf necrosis and drooping, and the stem end will have brown necrotic streaks. The plant grows only halfway or is completely dwarfed and wilted, and causes chlorotic rings on the green fruit [7].

The *Sw-5* gene is a single dominant quality of resistance gene that protects a wide range of Tospoviruses [8], originating from *L. peruvianum*, and has been identified and introgressed in the fresh tomato cultivar (*Lycopersicon esculentum*). *Sw-5* was discovered near the telomeric area of chromosome 9, between the CT71 and CT220 RFLP markers [9], and more closely linked to the CT220 marker (about within 65 kb) [10]. *Sw-5* locus is part of a loosely clustered gene family that includes six homologous paralog genes: *Sw-5a*, *Sw-5b*, *Sw-5c*, *Sw-5d*, *Sw-5e*, and *Sw-5f* are all potential variations [11]. *Sw-5a* and *Sw-5b* genes are strongly homologous (95%), and only *Sw-5b* has universal various isolates of TSWV [12]. *Sw-5b* was also linked to Tospovirus resistance to TCSV (tomato chlorotic spot virus), TZSV (tomato zonate spot virus), and GRSV (groundnut ringspot virus) [10].

TSWV is an RNA globular virus with an envelope structure and an outer membrane with a continuous protrusion layer. Its genome consists of three separate genomic RNA strands: long (L) RNA, medium (M) RNA, and short (S) RNA, all of which code for five different proteins. RdRp (RNA-dependent RNA polymerase) is a replication-related protein that interacts with host encoding factors. L RNA is a negative-sense RNA that encodes RdRp (RNA-dependent RNA polymerase) [13]. M RNA is an antisense RNA that codes for the amino acid and carboxy-terminal location of the glycoprotein precursor (GnGc), which is required for virions to assemble, mature, and release into their host [14]. The viral nonstructural proteins (NSm), which are encoded by sense RNA, are the main promoters of TSWV infection [15]. S RNA also contains a double sense RNA, with antisense RNA encoding nucleocapsid proteins (N) and sense RNA encoding nonstructural proteins (NSs) [16]; both proteins have a crucial function in the TSWV infection cycle [17]. Tomato spotted wilt virus (TSWV) is one of the most important plant viruses in the world [18]. TSWV belongs to the species Tospovirus that infect only plants, the genus *Orthotospovirus*, the family *Tospoviridae*, and the order *Bunyaviridae* [19]. TSWV caused widespread reductions in tomato yield and commercial value, together with plant mortality. Furthermore, the fast proliferation of TSWV-carrying *Frankliniella occidentalis* (Western flower thrips) has affected tomato agriculture and output severely [6]. As a result, tomato faces a major threat from this virus. We speculated on which genes are used to defend this virus, and considered the utilization of resistant/tolerant tomato cultivars to remedy the problem. These findings may lead to a better understanding of how the PR-10 protein and the *Sw-5b* gene act in the generation of systemic necrosis by the tomato spotted wilt tospovirus, which could aid in the future development of new antiviral approaches.

## 2. Results

### 2.1. Genome-Wide Identification and Analysis of Solanum Lycopersicum PR-10 Genes

From a genome-wide investigation of tomatoes, 45 putative *PR-10* encoding gene candidates were discovered. There are no chromosomal locations for the two *PR-10* gene ID numbers. Table 1 contains basic information on the *PR-10* genes, such as protein sequence length, isoelectric points, and molecular weight. PR-10 proteins varied in length from 64 to 210 amino acids, with molecular weights ranging from 7.6 to 24.4 kDa. The bulk of PR-10 proteins was classified as acidic based on theoretical isoelectric point (pI) values. The severe acidic or basic characteristics of each *PR-10* gene may contribute to different activities. In addition, all of the PR-10 protein sequences in tomatoes were shown to have the CDS, molecular weight, pI, instability index, aliphatic index, GRAVY, and subcellular localization (Table 1).

### 2.2. Phylogenetic Analysis, Conserved Motifs, and Gene Structures

To further characterize and identify potential functional relationships between the PR-10 proteins of tomato, a phylogenetic tree was constructed and 45 gene IDs of PR-10 protein were identified (Figure 1). The neighbor-joining method, with 1000 bootstrap reconstruction and completed deletion gaps/missing data, yielded five known subfamilies.

A total of nine conserved motifs were identified using the MEME server (Figure 2), and only motif 1 and motif 6 were found to be associated with the Bet_ V_ 1 (PF000407) domain. These two motifs (motif 1 and 6) were found in all SlPR-10 proteins, as expected. Remarkably, motif 9 was found only in the Solyc03g117450.1 and Solyc03g117460.1 protein sequence. The fewest motifs were identified as in Solyc01g011470.1 and Solyc01g081130.2 protein sequences. These variations of the protein motifs in number, types, and positions may be related to molecular function and roles in different tomato metabolic pathways.

The distribution of the intronic phase and the location of exons/introns are key features for gene structure study. Introns are divided into three phases: phase 0 introns are found between two consecutive codons, phase 1 introns are found between a codon’s first and second nucleotide, and phase 2 introns are found between a codon’s second and third nucleotide [20]. The exon–intron and intron phase organization of tomato *PR-10* genes is shown in Figure 3. The location and length of introns–exons organization in tomato *PR-10* genes are shown in Table 1.

### 2.3. Chromosome Mapping of the PR-10 Gene

CDD (Conserved domains database) from NCBI was used to compare the domain architecture of *PR-10* gene sequences. This study’s findings include the presence of conserved domains in their sequences, indicating that they are homologs. In silico chromosomal mapping of tomato *PR-10* genes are shown in Figure 4. Tomato *PR-10* genes have been discovered on ten of the twelve chromosomes. There are 11 genes on chromosome 4 and 13 genes on chromosome 9 in tomatoes; however, PR-10 genes were not discovered on chromosomes 2 and 11.

### 2.4. GO Analysis

Six biological processes and molecular activities and nine cellular components were identified via a gene ontology (GO) study, shown in Figure 5. Most PR-10 genes have a role in defensive responses and a reaction to abiotic stimulus in a biological function, according to the GO enrichment study. It exhibited protein kinase activity together with adenyl nucleotide and purine ribonucleoside binding activities at the molecular level. In the extracellular area, the cellular component had a function (Table 2).

### 2.5. Three-Dimensional Structure of the PR-10 Protein

There was a difference in the percentage of structural attributes that changed among the 45 *PR-10* gene IDs. Despite the 3D structural variances revealed among tomato PR-10 proteins, all 45 candidates have the binding proteins essential for protein interaction. For *PR-10* gene cloning and plant transformation, we chose the Solyc09g090980 gene ID of PR-10 protein, which is a phenolic, oxidative coupling protein with a length of 1424 bp and is situated on chromosome 9, as shown in Figure 6.

### 2.6. Transcriptomic Analysis of the PR-10 Protein of Tomato

The root, stem, branch, tendon, male flower, female flower, and fully opened young leaf were all subjected to spatial expression profiling of all *PR-10* genes in tomatoes. The *PR-10* genes were expressed differently in different plant tissues, according to the heatmap analysis. However, most of the genes in tomatoes had a relatively low expression level. Furthermore, the highest level of PR-10 expression of the Solyc09g090980 gene compared to the other tissues (Figure 7).

### 2.7. Incidence and Disease Index Level of TSWV in Each Group of Tomato Plants at Each Observation Time Point

Tomato-sensitive cultivar M82 and TSWV-resistant cultivar H8 were evaluated for their response to TSWV infections. As previously stated, plants were manually infected with TSWV and kept in an insect-proof greenhouse [21]. M82 and H8 plants treated plants with TSWV virus inoculum compared with M82 and H8 CK plants. It was found that there were no obvious symptoms of TSWV after 7 days of inoculation. After 14 days, the plants showed obvious dwarfing, and the growing point of the plant was purple; all M82 treatment groups showed purple spots in growth and wilting after 21 days. Incidence of TSWV in each CK and treatment (TR) group of M82 tomato plants showed the highest TSWV incidence rate compared to H8 plants (Table 3) after 28 days.

The severity of TSWV symptoms was scored on a scale of 0–4 (Table 4) for each plant at 14, 21 and 28 days post inoculation (dpi) (Table 3) to calculate the TSWV disease incidence rate and disease index level.

On the other hand, the plant disease index level of TSWV in each group of CK and TR; M82 showed the highest sensitivity compared to H8 plants, and H8 plants showed resistance phenotype after 28 days of infection of TSWV (Table 3).

After inoculation of TSWV, M82 and H8 plants are (shown in Figure 8) TSWV disease severity showed after 28 days of infection; the plant’s leaves were purple and dark brown patches in M82 plants (shown in Figure 9).

### 2.8. PR-10 Expression in M82 and H8 Plants

The purpose of this study was to determine how PR-10 protein expression affected infection in M82 and H8 plants following TSWV virus inoculation. Both CK and TR plants were sampled at various time intervals, including 0, 3, 6, 12, 24, 48, 72, 96 h and 7, 14, 21, 28 days. The expression pattern of PR-10 in different tissues of leaves of the tomato M82 (sensitive) and H8 (resistant) plants after the TSWV virus inoculation follows: M82 inoculated plants showed a wide range of response of PR-10 protein (5.70-fold relative expression at 72 h) compared to CK and H8 inoculated plants, which showed a wide range of response of PR-10 protein relative expression compared to CK (6.74-fold at 24 h) and significantly upregulated in the leaves after the TSWV inoculation (Figure 10).

### 2.9. TSWV-cp Expression in M82 and H8 Plants

To examine the effect of TSWV-cp protein expression after the TSWV virus inoculation in M82 and H8 plants. Leaf samples were taken at different time points such as 0, 3, 6, 12, 24, 48, 72, 96 h and 7, 14, 21, 28 days both CK and TR plants. The expression pattern of TSWV-cp in different tissues of leaves in the tomato M82 and H8 plants after the TSWV virus inoculation, M82 infected plants showed a wide range of response of TSWV-cp compared to CK (2.22-fold relative expression at 14D) were significantly upregulated, and H8 infected plants showed a narrow range of response of TSWV-cp compared to CK were non-significantly upregulated in leaves after the TSWV inoculation in Figure 11.

### 2.10. Expression of Sw-5b in M82 and H8 Plants

The purpose of this study was to determine how *Sw-5b* expression changed following TSWV virus inoculation in M82 and H8 plants. Tomato leaf samples were taken from CK and TR plants at various times, including 0, 3, 6, 12, 24, 48, 72, 96 h and 7, 14, 21, 28 days. Figure 12 shows the *Sw-5b* expression pattern in different tissues of tomato M82 and H8 plants’ leaves after TSWV virus inoculation; M82-infected plants showed no significant expression of *Sw-5b*, while H8-infected plants showed a wide range of relative expression of *Sw-5b* (4.14-fold at 7D) and significantly upregulated in leaves after TSWV inoculation.

### 2.11. Expression of Different Index Levels of Leaves in M82 and H8 Plants

The purpose of this study was to determine how PR-10, TSWV-cp, and *Sw-5b* expression affected different disease index levels of M82 and H8 tomato leaves following TSWV inoculation. After 28 days, tomato leaf samples were obtained from both CK and TR plant leaves. In the Section 4, different disease index levels are explained. Different disease index levels show various types of expression of PR-10, TSWV-cp, and *Sw-5b* in M82 and H8 plants. H8-infected plants showed a higher range of PR-10 protein compared to M82 plants. On the other hand, M82-infected plants showed a wide range of TSWV-cp (4.02-fold relative expression at level 3) compared to H8 plants. Additionally, M82 plants showed no significant expression of *Sw-5b*, but H8 plants showed 3.53-fold relative expression of *Sw-5b*, as shown in Figure 13.

### 2.12. Expression of Different Index Levels of the Tomato Plants Root

To examine the effect of PR-10, TSWV-cp, and *Sw-5b* expression of different index levels of M82 and H8 tomato plant roots, tomato plant root samples were taken after 28 days of CK and TR plants. Different index levels show various types of expression levels of PR-10, TSWV-cp, and *Sw-5b* in M82 and H8 plants. H8-infected plants showed a higher range of PR-10 protein compared to M82 plants. Additionally, M82-infected plants showed a wide range of response of TSWV-cp (4.72-fold relative expression at level 2) and significantly upregulated, and H8 plants showed relatively lower expression of TSWV-cp. M82 plants also showed no significant expression of *Sw-5b*, but H8 plants showed 3.40-fold relative expression that was significantly upregulated (Figure 14).

### 2.13. Correlation between PR-10 and Sw-5b Gene Expression after TSWV Inoculation

Table 5 shows the results of a correlation study between the PR-10 and *Sw-5b* gene expressions following TSWV inoculation. At multiple time intervals, such as 3, 6, 12, 24, 48, 72, 96 h and 7, 14, 21, 28 days, the studies revealed extremely significant correlations between PR-10 and *Sw-5b* genes. M82 and H8 tomato plants exhibited extremely significant correlations in this study; however, H8 plants showed significant correlations when compared to M82 plants.

### 2.14. Heat Map of PR-10 Protein and Sw-5b Gene Expression at Different Time Points

PR-10 protein and *Sw-5b* gene expression at various time periods following TSWV inoculation. TB tools software was used to create the heat map, which was then applied to two clades of M82 and H8 plants. Figure 15 shows that PR-10 protein and *Sw-5b* gene expression patterns were clustered; H8 plants demonstrated resistance to TSWV of various colors with high PR-10 protein and *Sw-5b* gene expression, whereas M82 plants showed susceptibility to TSWV. We have a comprehensive understanding of the PR-10 protein and *Sw-5b*, both of which play an equal role in TSWV resistance following TSWV inoculation. H8 plants were more resistant to *Sw-5b* than M82 plants because M82 plants lack *Sw-5b* resistant genes, but H8 plants have the resistant genes.

## 3. Discussion

To confirm that the PR-10 proteins found in tomatoes are related to pathogenesis-related proteins from the PR-10 family, a general NCBI-BLAST was performed using their deduced amino acid (a.a) sequence as the query of FASTA search. PR-10 protein sequences are near in molecular weight (17–18 kDa) and amino acid length (157–166 a.a) with a pI value of 4.69–6.17 and a Bet_ V_ 1 (PF000407) domain structure. The subcellular localization of the PR-10 protein was predicted to be cytoplasmic (Table 1).

To characterize and identify potential functional relationships between the PR-10 proteins of tomato, a phylogenetic tree was constructed and 45 gene IDs of PR-10 protein identified. The neighbor-joining method, with 1000 bootstrap reconstruction and completed deletion gaps/missing data, yielded five known subfamilies. The subsequent phylogenetic analysis of all forty-five PR-10 nucleotide and protein sequences were carried out using the maximum-likelihood method with 1000 bootstraps (Figure 1).

Although PR-10 proteins are mostly recognized for their roles in plant defense in response to biotic and abiotic stress, the overall biological function of many PR-10 members are yet unknown. We investigated the *PR-10* genes to better understand the evolutionary relationship between tomato genes and proteins. The findings of this work serve as a theoretical foundation for future research into the structural relationships that define pathogenesis-related proteins [22]. To fight broad pathogenic activity, plants have evolved several defensive mechanisms. Among other strategies, they create antibiotic chemicals such as phytoalexins [23]; and multiple pathogens such as viruses, bacteria, and fungi induce the expression of several genes, as well as the formation of ethylene and salicylic acid in the plant after pathogen infection, resulting in a stress response [24]. The induced genes inform the plants to generate pathogenesis-related (PR) proteins, which are involved in a defensive mechanism [25]. The level at which new cases of a disease appear in a plant population within a given period is referred to as incidence and the disease intensity is calculated using an interval scale measure, which has been used to quantify a disease severity index (DSI) on a percentage basis [26]. We measured different types of diseased leaf samples at different time points and estimated disease incidence and index level (Table 3) with the information contained in Table 4.

The PR-10 family is a critical element of pathogenesis-related proteins, which aid plant defense against biotic and abiotic stress. For example, several tobacco PRs have been identified as chitinases and β-1-3 glucanases with antifungal activity [27]. Multigene families code for PR-10 proteins. This may be the reason for the multifunctional existence of these ancient proteins, which spent time in a phase known as protein promiscuity for mutations and functions [28]. The cAMP-dependent protein kinase, casein kinase II, and protein kinase C also have phosphorylation sites on several of the PR-10 proteins [29].

Tomato spotted wilt virus (TSWV) is transmitted by thrips and has become one of the most important viral vector complexes for agriculture and food defense [30]. One of the most effective approaches to decrease viral diseases is to grow virus resistant cultivars. This technique’s effectiveness is dependent on the presence of resistant genes in either cultivated or wild relatives. The most prevalent mechanism of natural plant resistance to viral infection is the hypersensitive response [31]. HR causes viral invasion-associated cells to die quickly, reducing viral cell-to-cell multiplication and the virus’s subsequent spread throughout the plant. The hypersensitive reaction is caused by the virus’s exact identification, which is based on comparable plant and viral gene products (HR). In Tospoviruses, particularly TSWV, the dominant genes of *Sw-5* are the principal sources of HR-based resistance [30]. *Sw-5* clustered proteins were discovered to be part of the resistance (R) gene family, encoded by the amino-terminal *Solanaceae* domain (SD) and coiled-coil domain (CC), a core nucleotide binding-adapter shared by APAF-1, R proteins, and a CED-4 (NB-ARC), and a leucine-rich repeat (LRR) domain [32]. Multiple homologs have been found in the tomato genome; nevertheless, *Sw-5b* provides broad and long-lasting resistance and has been extensively researched because of this functionality [11]. The NSm encoded in the TSWV M segment is the TSWV product that triggers the resistance response (*Avr* Determinant) of *Sw-5b*-mediated resistance [11,32,33]. *Sw-5* containing tomato cultivar have L and S segments, as well as the M segment form a TSWV resistance-inducing (RI) isolate and a TSWV resistance-breaking (RB) isolate [34]. Because H8 plants had *Sw-5b* resistant genes, PR-10 protein expression was higher in H8 plants compared to M82 plants (Figure 10); nevertheless, TSWV-cp protein expression was lower in H8 plants, as seen in Figure 11. However, due to these resistance genes (*Sw-5b*) (Figure 12), *Sw-5b* was substantially expressed in H8 plants.

The viral small interfering RNAs (vsiRNAs) originating from the TSWV genome have been discovered to potentially target host genes involved in a variety of processes, including plant–pathogen interactions [35]. As a result, they experience much plant protection and autoimmunity. Plant pattern-recognition receptors (PRRs) sense the virus that has infected the plants by recognizing pathogen-associated molecular patterns (PAMPs). The plants’ first line of defense against an immunological response is PAMP-triggered immunity (PTI). On the other hand, rapid pathogen effectors hinder the PTI response. PAMP-triggered immunity (PTI) and effector-triggered immunity (ETI) are two types of immune systems found in plants. PTI is elicited by pathogen/microbe-associated molecular patterns (PAMPs/MAMPs), while ETI is elicited by disease resistance plant proteins (R), which are highly effective for disease resistance reactions upon precise identification of pathogen effectors, and both PTI and ETI cause local immune responses. During virus invasion, nucleotide-binding leucine-rich repeat receptors (NLRs) recognize specific pathogen effectors and induce effector-triggered immunity (ETI), which triggers a long-distance protective mechanism known as SAR (systemic acquired resistance) in plants [36].

NLRs (nucleotide-binding leucine-rich repeat receptors) identify unique pathogen effectors during virus invasion and cause effector-triggered immunity (ETI) in the plant [37]. Because of their distinct N-terminal configurations, plant NLRs are known as coiled–coil (CC)-NLRs (CNLs) or Toll/interleukin-1 (TIR)-NLRs (TNLs) [38]. The NLRs specifically or implicitly recognize pathogen effectors and initiate a hypersensitive cell death response (HR) aimed at limiting TSWV infection to the site of infection [39]. TSWV-cp protein was highly expressed in M82 sensitive plants, but PR-10 protein and *Sw-5b* were highly expressed in H8 plants leave and roots in different types of disease index levels (Figure 13 and Figure 14), and PR-10 protein and *Sw-5b* are highly correlated for TSWV disease resistance in the plants (Figure 15). The tomato *Sw-5b* belongs to the coiled–coil leucine-rich repeats (CNLs) and also auto-activated. *Sw-5b* is a typical class of proteins, CC-NB-ARC protein, and *Sw-5b* has broad-spectrum resistance that its SD domain can specifically recognize a 21- amino acid (21-aa) in NSm of TSWV. The region of NSm is highly conserved in the American-type Tospoviruses only, and not in Euro-Asian-type Tospoviruses [39]. At the moment, the *Sw-5b* genes have been discovered in various tomato germplasm materials by genome sequencing, and the sequence variations between the genes of different tomato plants are very large [40]. TSWV-specific NSm attaches to the expanded SD domain of the *Sw-5b* protein, activating the switch that triggers the receptor and HR, resulting in a vigorous defensive reaction against Tospoviruses [40]. The CNLs (including *Sw-5b* protein) from tomato, on the other hand, detect NSm and NSs with robust effector-triggered immunity (ETI) and activate the hypersensitive cell death reaction (HR). Salicylic acid (SA), jasmonic acid (JA), and ethylene (ET) play critical roles in PTI and ETI immunity, along with helping plants to establish systemic acquired resistance (SAR) [41]. The SA signaling pathway is essential in tomato plants for basal protection against TSWV [42]. The SA accumulates in the contaminated regions, where it then triggers the rapid activation of transcriptional resistance (R) genes [43]. These findings suggest that PR-10 protein and its involvement with *Sw-5b* have a resistance mechanism against TSWV to protect against these devastating viruses in tomatoes, which also establish systemic acquired resistance (SAR) in the plants.

## 4. Materials and Methods

### 4.1. Materials

Forty-five amino acid sequences of PR-10 encoding genes were used to analyze the genome-wide identification of the PR-10 superfamily. M82 and H8 tomato seeds were used in this experiment as a sensitive and resistant material because M82 was included as a TSWV susceptible cultivar, as reported by [44], and H8 was included as a TSWV resistant cultivar, as reported by the Laboratory of Tomato Quality and Stress Tolerance Regulation Mechanism and Genetic Improvement, College of Horticulture, Northwest A and F University, China.

### 4.2. Methods

#### 4.2.1. Identification and Sequence Analysis

The PR-10 protein amino acid sequences were used as a reference point for a number of database searches against the Phytozome database proteome files. (https://phytozome-next.jgi.doe.gov/) accessed on 23 May 2021. The National Center for Biotechnology Information (NCBI) provided isolated versions of BLASTP, which were utilized with an e-value threshold of 1e^−10^ [45]. The candidate sequences were further screened by searching for the PR-10 domain in tomato by using PFAM (http://pfam.xfam.org/) accessed on 23 May 2021 and SMART (http://smart.embl-heidelberg.de/) accessed on 23 May 2021. The characteristic of PR-10 protein, including protein length, isoelectric point (pI), molecular weight (*M*_W_), and grand average of hydropathicity (GRAVY) were predicted by ExPASy ProtParam (http://web.expasy.org/protparam/) accessed on 23 May 2021. The signal peptides and transmembrane (TM) domains were predicted with SignalP 4.1 (http://www.cbs.dtu.dk/services/SignalP/) accessed on 23 May 2021.

#### 4.2.2. Phylogenetic Analysis

To compare the evolutionary relationships and identify the subfamilies of PR-10, proteins were used to construct the phylogenetic tree using MEGA-X with the neighbor-joining (NJ) method [46]. The phylogenetic tree was then visualized by iTol (https://itol.embl.de/) accessed on 24 May 2021.

#### 4.2.3. Gene Structure, Conserved Motifs, and Chromosome Mapping Analysis

Gene Structure Display Server 2.0 software (http://gsds.cbi.pku.edu.cn/) accessed on 23 May 2021 was used to investigate the exon–intron organizations of PR-10 genes based on their information given in the Phytozome database. The novel motifs of PR proteins were searched using MEME 5.0.3 (http://meme-suite.org/tools/meme) accessed on 23 May 2021. The parameters were set as follows: the site distribution was any number of repetitions (anr); the number of motifs was 20; the width of the motif was limited to between 10 and 30, and other optional parameters remained as the default. The combination of gene structures, motifs, and the phylogenetic tree was then generated using the iTol tool. The distributions of *PR-10* genes on tomato chromosome mapping were illustrated with MapInspect 1.0 (http://mapinspect.software.informer.com/) accessed on 24 May 2021.

#### 4.2.4. GO Analysis

Biological processes, cellular components, and molecular function are the three categories of gene ontology (GO). Using the PANNZER2 web server, the *PR-10* genes were investigated for their role in GO (http://ekhidna2.biocenter.helsinki.fi/sanspanz/) accessed on 23 May 2021.

#### 4.2.5. Three-Dimensional Structure of the PR-10 Protein

A Protein Homology/Analogy Recognition Engine v2 (Phyre2) server was used to generate the anticipated 3D structures. (http://www.sbg.bio.ic.ac.uk/~phyre2/html/page.cgi?id=index) accessed on 23 May 2021.

#### 4.2.6. Transcriptomic Analysis

Genome-wide expression data from tomatoes were utilized to reveal the expression patterns of the PR-10 protein family in different tissues and developmental periods. Transcriptomic data from several different plant tissues, including leaves, stems, roots, and various developmental stages of tomato fruit, were downloaded from the Sol genomics network (https://solgenomics.net/) accessed on 30 May 2021, and then TB tools software was used to analyze the expression levels of *PR-10* genes in various tissues of tomato plants by using 45 gene IDs of PR-10 protein.

#### 4.2.7. TSWV Virus Solution Stock Preparation

TSWV virus solution stock was prepared according to the method of [30]. A 0.5 g diseased leaf sample was ground in a mortar and pestle and mixed with a buffer solution (pH 7.0) of 10 mL (0.1 mol/LH24 PO_4_)/Na_2_HPO_4_; 2% poly-vinyl pyrolidon; and 0.2% Na_2_SO_3_. The TSWV virus solution was kept at −80 °C prior to further analysis.

#### 4.2.8. Plant Materials and Growth Conditions

To minimize pests and disease incidence during seedling growth and development, the experiment was conducted in the greenhouse at the College of Horticulture Northwest Agriculture and Forestry University in Yangling, China. The seeds were germinated for 3 days on moist filter paper in Petri dishes at 28 °C in the dark. The seedlings were transported to a growth chamber with a 16 h light/8 h dark photoperiod and a temperature cycle of 25/16 °C. The plants were moved to a solar greenhouse at Northwest A and F University in Yangling, Shaanxi Province, China, at the four true-leaf stage. The plants were cultivated under a 22/18 °C day/night temperature during the TSWV inoculation.

#### 4.2.9. Virus Inoculation

The TSWV virus was inoculated into six-week-old seedlings. A syringe was used to inject the TSWV virus. The injections were made with a 1.0 mL syringe at three sites on the leaf: border, middle, and base, covering around 0.5 cm^2^ of leaf area. The presence of the virus was determined visually and confirmed by PCR.

#### 4.2.10. Treatment and Sampling

Inoculated treatment (TR) and non-inoculated (CK) seedlings were used to split tomato cultivars into two groups. At 0, 3, 6, 12, 24, 48, 72, 96 h, and 7, 14, 21, 28 days, samples were taken from the control and treatment groups. TSWV infection of tomato leaves was identified by PCR using TSWV-_CP_ specific primers. At 14, 21, and 28 days after inoculation, the incidence and disease index of TSWV were evaluated, along with the disease symptoms of susceptible and resistant plants, to determine the relative expression of TSWV-cp, PR-10, and *Sw-5b*. There were 20 plants in each treatment. For this experiment, three biological replicates were used. Viral disease classification standard records disease index level in Table 4 [47].

#### 4.2.11. Investigation of Incidence Rate and Disease Index

The incidence of TSWV was observed at 14, 21, and 28 days after inoculation, and the disease symptoms of susceptible and resistant plants were observed.

Incidence rate (%) = (number of infected plants / total number of treated plants) × 100.

Measurement of Disease index = (number of disease plants at all levels × number of representatives at all levels)/(total number of investigated plants × maximum series)] × 100 (at 14, 21, and 28 days after inoculation).

#### 4.2.12. RNA Extraction and qRT-PCR

Leaf samples were collected from three TSWV-inoculated and three uninoculated plants. The leaves were directly frozen in liquid nitrogen and kept at −80 °C for further analysis. Total RNA was derived from tomato leaves using an Omega plant RNA kit, and cDNA was generated using an M-MuLV reverse transcriptase kit (Thermo Scientific, USA). For TSWV-cp protein confirmation, the cDNA samples were amplified using PCR: 95 °C for 5 min, 35 cycles of 95 °C for 30 s, 53 °C for 30 s, 72 °C for 3.5 min, and 72 °C for 10 min. Another cDNA was also synthesized for qPCR using the Evo M-MLV RT Kit (Accurate Biotechnology, Hunan). PCR was used to amplify the cDNA samples: 42 °C for 2 min, 37 °C for 15 min, and 85 °C for 5 s. Thermocycler iQ5 Real-Time PCR Detection Device (BIO-RAD Corp., Hercules, CA, USA) and SYBR Green Pro Taq HS qPCR (Accurate Biotechnology, Hunan) were then used to operate qRT-PCR according to manufacturer guidance. The PCR conditions were as follows: 95 °C for 30 s, accompanied by 40 cycles of 95 °C for 5 s, 60 °C for 30 s, and 72 °C for 20 s. As an internal control, the tomato *actin* gene was used. The qRT-PCR data were analyzed using the 2−ΔΔ _CT_ method [48]. The primers are listed in Appendix A.

#### 4.2.13. Statistical Analysis

GraphPad Prism version 7.00 for Windows (one-way ANOVA followed by Dunnett test) GraphPad Software, San Diego, California, USA, www.graphpad.com, accessed on 23 May 2021, was used to plot and statistically analyze the graphs. A value of *p* > 0.05 was regarded as statistically significant; * *p* = 0.05, ** *p* = 0.01, *** *p* = 0.001, **** *p* = 0.0001, and ns = non-significant. Correlation coefficients were determined by SPSS 25.0 (IBM, Armonk, NY, USA).

## 5. Conclusions

We found 45 PR-10 protein superfamily candidate genes with lengths ranging from 64 to 210 amino acid residues and molecular weights ranging from 7.6 to 24.4 kDa. The 5 subfamilies of PR-10 protein was discovered, as well as 9 conserved motifs. On ten of the twelve chromosomes, *PR-10* genes have been found. Gene ontology research revealed six biological processes and molecular activities, as well as nine cellular components, as well as the greatest level of expression in the Solyc09g090980 *PR-10* gene compared to other tomato tissues. TSWV infection strongly induced *PR-10* and *Sw-5b* gene transcription and activity in tomato leaves, H8 plants having the highest significant expression of PR-10 at 24 h (6.74-fold) and *Sw-5b* gene at 7 days (4.14-fold) compared to control after the TSWV inoculation, TSWV inoculated M82 plants showed significant expression of *PR-10* gene at 72 h (5.70-fold) and no significant expression of *Sw-5b* gene all the time, compared to control. The expressions of *PR-10* and *Sw-5b* genes showed highly significant correlations in H8 plants after the inoculation of TSWV at different time points and also heat map showed that these two genes may participate in regulating the defense response after the inoculation of TSWV in tomatoes. Thus, we conclude that *PR-10* and *Sw-5b* genes have an important role in the defense response after the infection of TSWV.

## Figures and Tables

**Figure 1 ijms-23-01502-f001:**
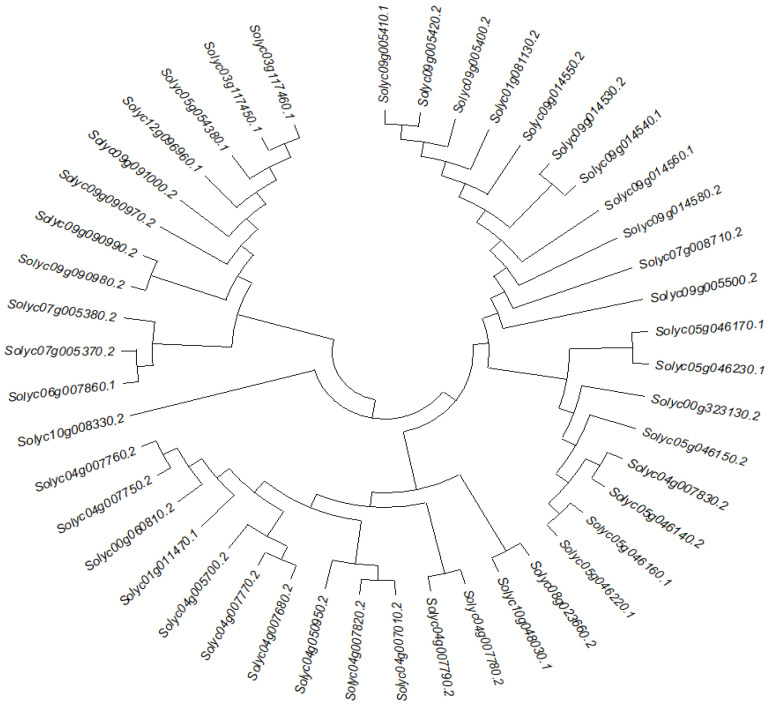
Phylogenetic tree of 45 genes of PR-10 protein in tomato. Sequence alignment was done by ClustalW and Phylogenetic tree generated by MEGA 5.1 with maximum likelihood (ML) method for 1000 replicates as the bootstrap value.

**Figure 2 ijms-23-01502-f002:**
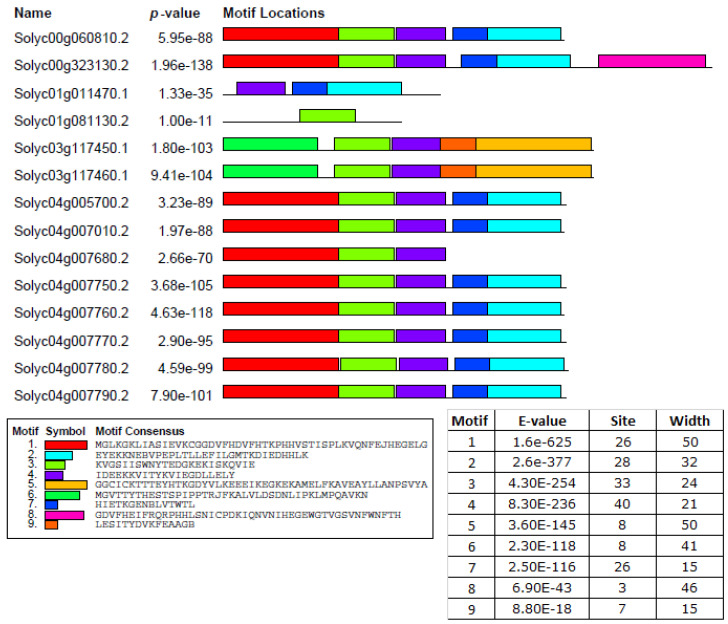
Motif locations of PR-10 protein. Each motif is represented using a box with different colors. Motif 1, red; Motif 2, violet; Motif 3, light green; Motif 4, light blue; Motif 5, yellow; Motif 6, green; Motif 7, blue; Motif 8, purple; Motif 9, orange.

**Figure 3 ijms-23-01502-f003:**
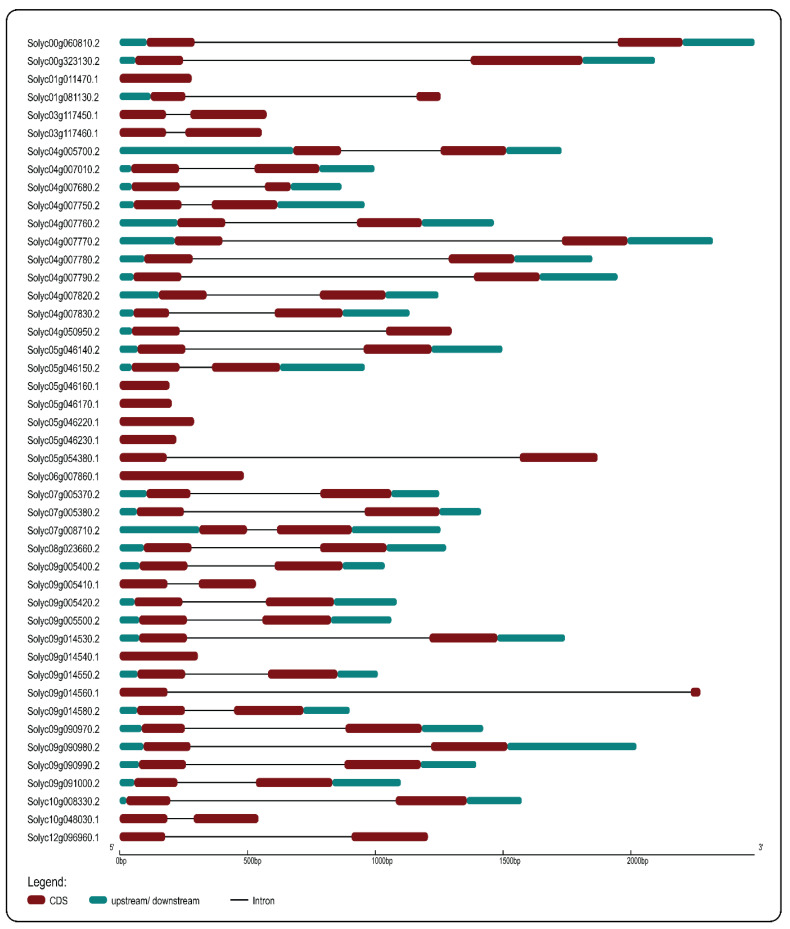
Gene structure of 45 genes of PR-10 protein in tomato. Exon, intron, upstream, and downstream regions are indicated as dark red boxes, black lines, and light blues boxes, respectively.

**Figure 4 ijms-23-01502-f004:**
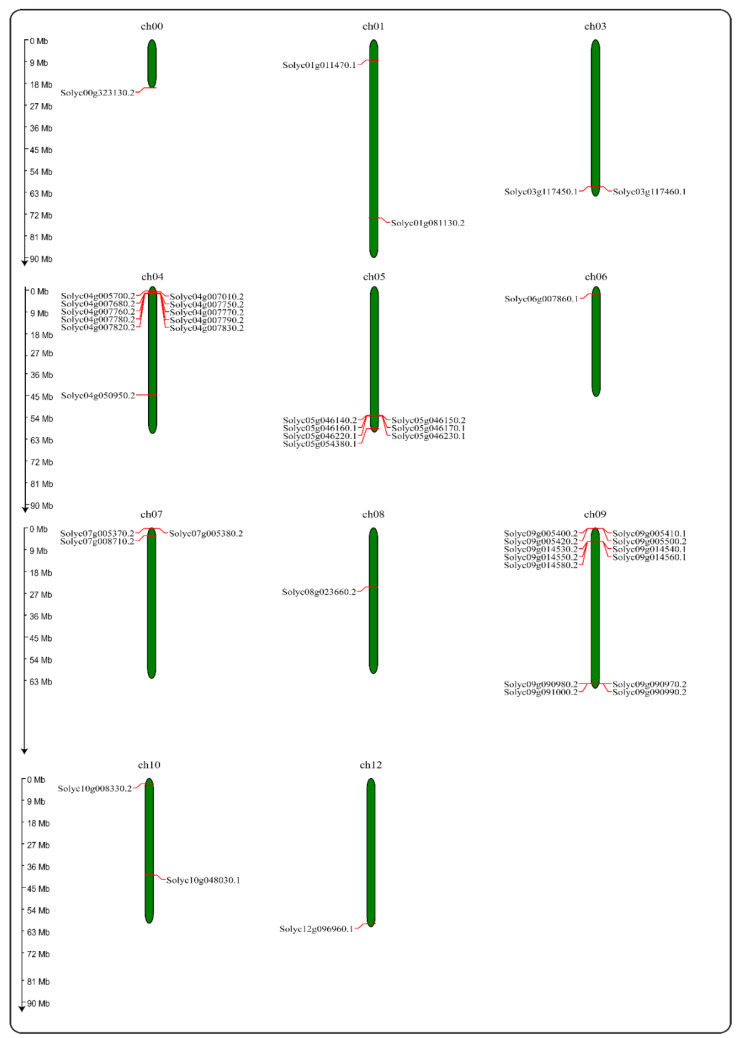
Chromosome mapping of the *PR-10* gene of tomato.

**Figure 5 ijms-23-01502-f005:**
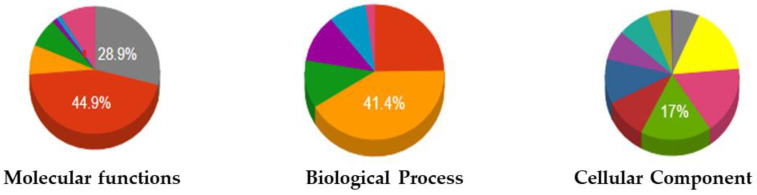
The gene ontology (GO) term distribution of PR-10 proteins of tomato.

**Figure 6 ijms-23-01502-f006:**
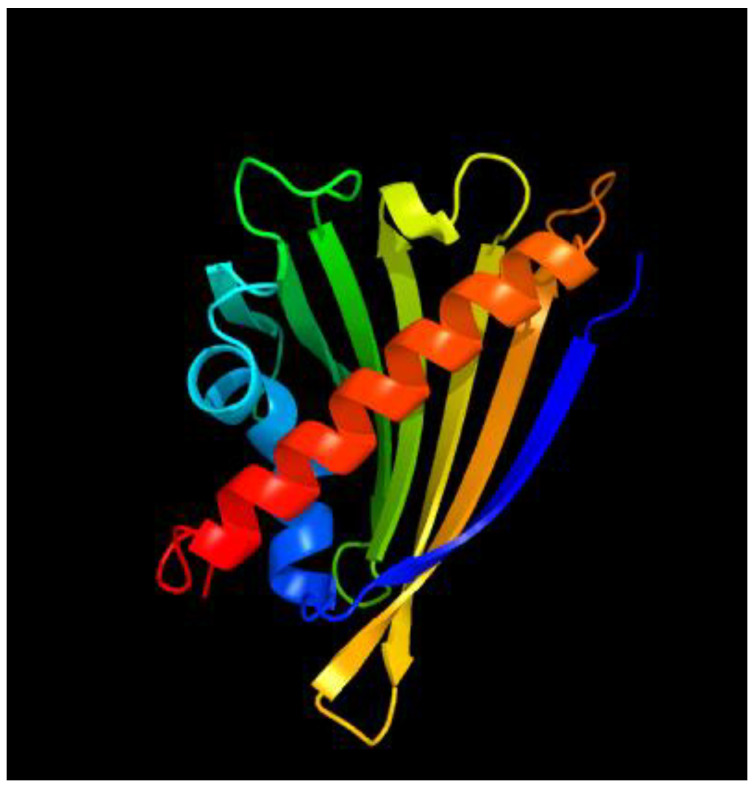
The predicted 3D structures of Solyc09g090980 of PR-10 protein, generated using the Phyre2 server, and binding pockets identified by the CASTp 3.0 server (shown in red).

**Figure 7 ijms-23-01502-f007:**
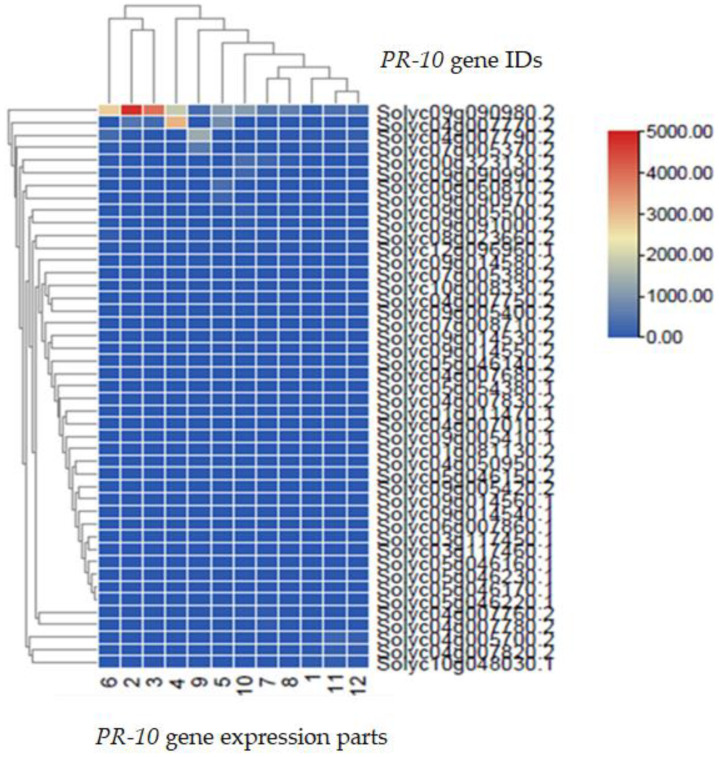
Transcriptomic analysis of the PR-10 protein in tomato. The data presented are based on at least three separate replicates. The gene expression within each family is represented using hierarchical clustering.

**Figure 8 ijms-23-01502-f008:**
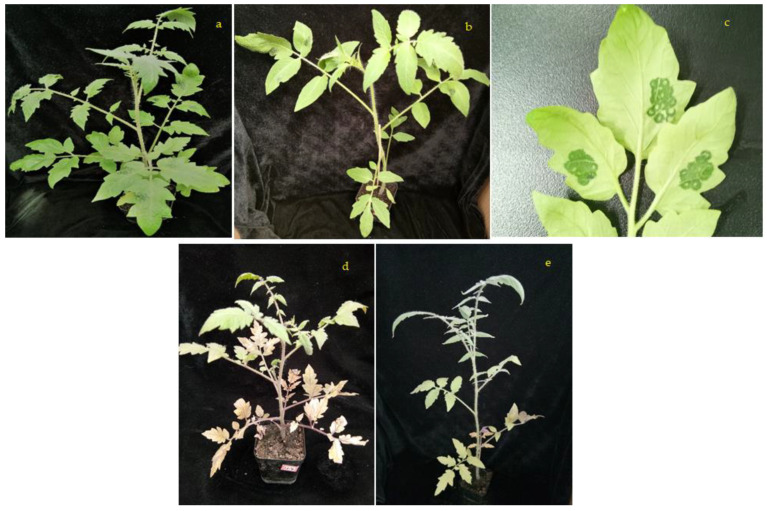
TSWV inoculate of M82 and H8 plants of tomato; (**a**) TSWV inoculation of M82 plant, (**b**) TSWV inoculation of H8 plant, (**c**) TSWV inoculated leaves, (**d**) TSWV inoculate of M82 plants after 28 days, and (**e**) TSWV inoculate of H8 plants after 28 days.

**Figure 9 ijms-23-01502-f009:**
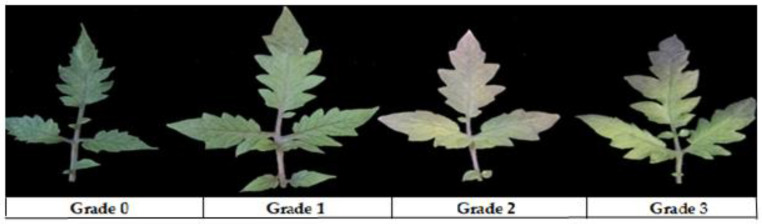
M82 plant leaves show typical symptoms of TSWV: Grade 0, no disease; Grade 1, slightly infected; Grade 2, 50% infected; and Grade 3, 75% infected leaves.

**Figure 10 ijms-23-01502-f010:**
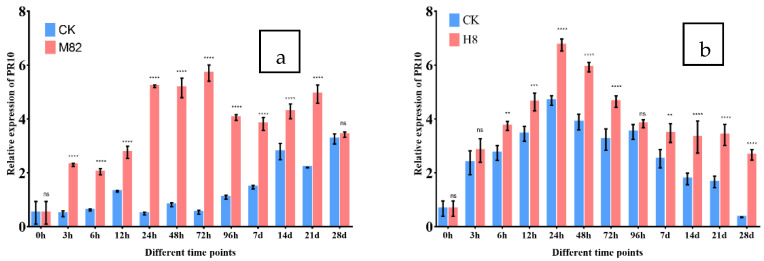
PR-10 expression in M82 and H8 plants. The relative transcription levels of TSWV-infected tomato M82 and H8 plants were normalized to the control gene *Actin*, as measured by qRT-PCR of PR-10 protein. At 0, 3, 6, 12, 24, 48, 72, and 96 h pi (hours post infection) and 7, 14, 21 and 28 dpi (days post infection), the expression levels in the leaf tissues were assessed (days post infection). The CK plants are shown by blue bars, while the M82 and H8 plants are represented by red bars. Three biological replicates’ mean results (±SD) are shown as relative changes. (**a**) PR-10 relative expression in M82 plants vs. CK (not inoculated with TSWV) and (**b**) PR-10 relative expression in H8 plants compared to CK. Asterisk indicates statistically significant; ** *p* = 0.01, *** *p* = 0.001, **** *p* = 0.0001, and ns = non-significant are the *p* values.

**Figure 11 ijms-23-01502-f011:**
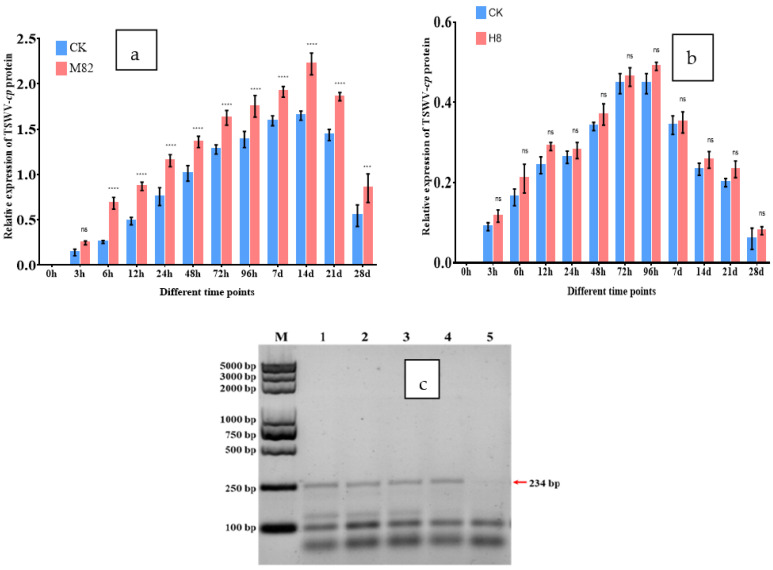
TSWV-cp expression in M82 and H8 plants. Tomato M82 and H8 plants infected with TSWV inoculum, and relative transcript levels normalized to the control gene *Actin.* At 0, 3, 6, 12, 24, 48, 72, and 96 hpi (hours post infection) and 7, 14, 21 and 28 dpi (days post infection), the expression levels in the leaf tissues were assessed (days post infection). The CK plants are shown by blue bars, while the M82 and H8 plants are represented by red bars. The mean values (±SD) from three biological replicates are provided as relative changes: (**a**) TSWV-cp relative expression in M82 plants compared to CK; (**b**) TSWV-cp relative expression in H8 plants compared to CK; and (**c**) TSWV-cp confirmation in Yangling, China, isolates 234 bp. Asterisk indicates statistically significant; *** *p* = 0.001, **** *p* = 0.0001, and ns = non-significant are the *p* values.

**Figure 12 ijms-23-01502-f012:**
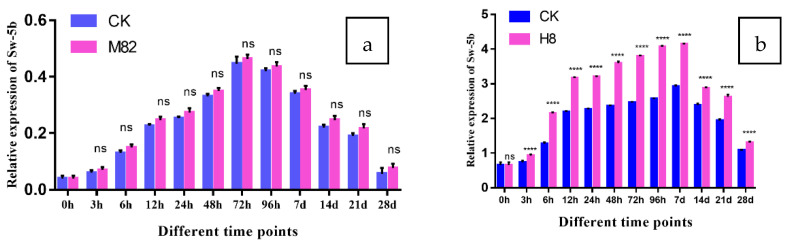
Expression of *Sw-5b* in M82 and H8 plants. *Sw-5b* relative transcription levels were assessed by qRT-PCR; tomato M82 and H8 plants were inoculated with TSWV, and relative transcript levels were normalized to the control gene *Actin*. At 0, 3, 6, 12, 24, 48, 72, and 96 hpi (hours post infection) and 7, 14, 21 and 28 dpi (days post infection), the expression levels in the leaf tissues were assessed (days post infection). The CK plants are shown by blue bars, while the M82 and H8 plants are represented by red bars. Three biological replicates’ mean results (±SD) are reported as relative changes. (**a**) *Sw-5b* expression in M82 plants in comparison to CK, and (**b**) *Sw-5b* expression in H8 plants in comparison to CK. Asterisk indicates statistically significant; **** *p* = 0.0001, and ns = non-significant are the *p* values.

**Figure 13 ijms-23-01502-f013:**
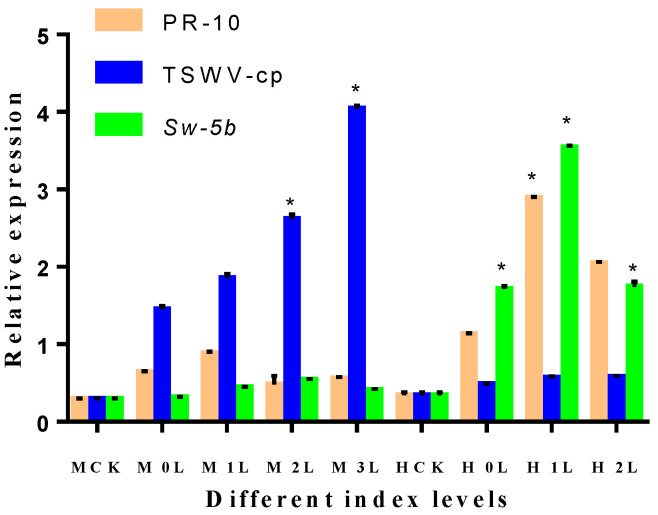
Expression of different index levels of leaves in M82 and H8 plants. After the TSWV inoculum was applied to tomato M82 and H8 plants, the relative transcript expression levels of PR-10 protein, TSWV-cp protein, and *Sw-5b* were determined using qRT-PCR, and the relative transcript levels were normalized to the control gene *Actin*. The expression levels in the leaf tissues were measured at different disease index levels. Light yellow bars represent the PR-10 proteins, blue bars represent the TSWV-cp protein, and light green bars represent the *Sw-5b*. The mean values (±SD) from three biological replicates are presented as relative changes. The least significant difference (LSD) test was used to determine if a gene showed significant induction as compared to the transcript abundance of the control group (MCK represents the M82, CK plant; MTR represents the M82-treated plant; HCK represents the H8, CK plant; and HTR represents the H8-treated plant). Asterisk indicates statistically significant; * *p* = 0.05, and ns = non-significant are the *p* values.

**Figure 14 ijms-23-01502-f014:**
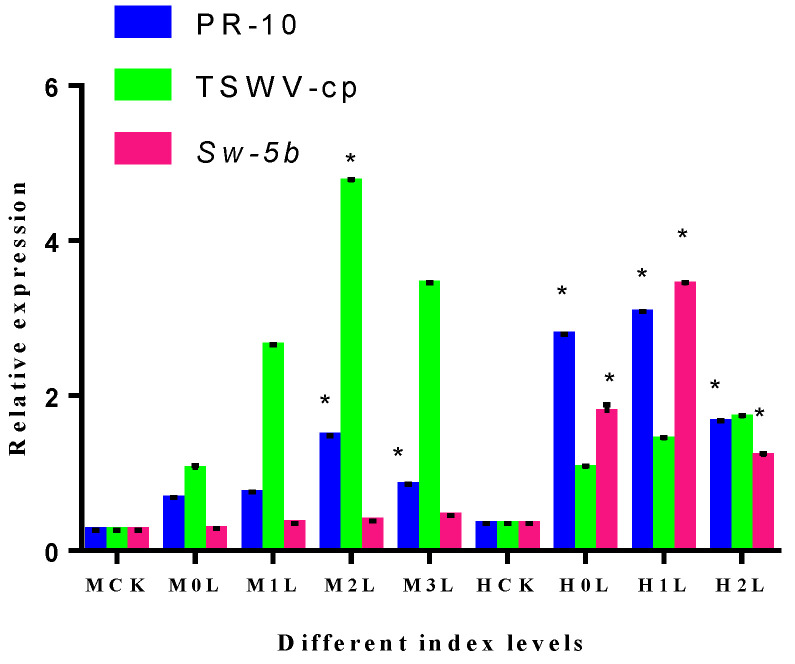
Expression of different index levels of plant roots in M82 and H8. Relative transcription levels determined by qRT-PCR of PR-10 protein, TSWV-cp protein, and *Sw-5b*; tomato M82 and H8 plants were infected with TSWV inoculum and the relative transcription levels were normalized to the control gene *Actin*. The expression levels in the root tissues were measured at different disease index levels. Blue bars represent the PR-10 proteins, light green bars represent the TSWV-cp protein, and purple bars represent *Sw-5b*. The mean values (±SD) from three biological replicates are presented as relative changes. The least significant difference (LSD) test was used to determine if a gene showed significant induction as compared to the transcript abundance of the control group (MCK represents the M82 CK plant; M represents the M82-treated plant; HCK represents the H8, CK plant; and H represents the H8-treated plant). Asterisk indicates statistically significant; * *p* = 0.05 and ns = non-significant are the *p* values.

**Figure 15 ijms-23-01502-f015:**
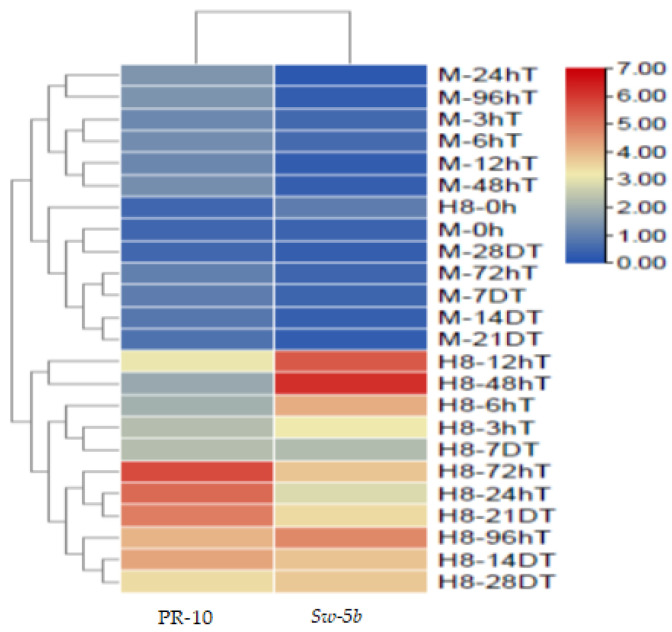
Heat map of PR-10 protein and *Sw-5b* at different time points after TSWV inoculation. *PR-10* and *Sw-5b* genes are highly expressed in H8 plants compared to M82 plants.

**Table 1 ijms-23-01502-t001:** The sequence features and physicochemical properties of tomato PR-10 proteins.

SL. No.	Chr	Start	END	Strand	Number of Amino Acids	CDS	Molecular Weight	pI	Instability Index	Aliphatic Index	GRAVY	Subcellular Localization
1	ch00	14,149,228	14,151,711	reverse	158	441	18,196.91	5.03	38.70	99.94	−0.218	Cytoplasmic
2	ch00	21,749,238	21,751,331	forward	210	633	24,401.45	5.15	43.03	86.62	−0.417	Cytoplasmic
3	ch01	9,342,757	9,343,038	reverse	93	282	11,233.13	7.65	50.57	90.11	−0.134	Cytoplasmic
4	ch01	80,454,898	80,456,153	forward	76	231	8763.32	6.71	30.32	86.97	−0.051	Cytoplasmic
5	ch03	66,611,609	66,612,183	forward	159	480	17,608.98	5.02	32.81	82.20	−0.35	Cytoplasmic
6	ch03	66,614,237	66,614,792	forward	159	480	17,636.95	4.90	34.90	79.12	−0.401	Cytoplasmic
7	ch04	455,936	457,664	reverse	147	444	16,774.40	6.29	35.91	86.19	−0.334	Cytoplasmic
8	ch04	737,430	738,426	forward	146	441	16,613.22	5.96	33.78	99.38	−0.168	Cytoplasmic
9	ch04	1,363,357	1,364,224	forward	95	288	10,838.61	9.64	21.65	109.58	−0.226	Cytoplasmic
10	ch04	1,429,473	1,430,430	forward	147	444	16,734.98	5.59	31.71	94.76	−0.318	Cytoplasmic
11	ch04	1,435,905	1,437,368	forward	146	441	16,583.08	6.03	25.97	98.01	−0.2	Cytoplasmic
12	ch04	1,456,588	1,458,908	reverse	147	444	16,595.16	5.96	42.21	106.05	−0.162	Cytoplasmic
13	ch04	1,466,509	1,468,357	forward	148	447	17,017.30	5.62	19.96	90.88	−0.403	Cytoplasmic
14	ch04	1,474,832	1,476,779	forward	147	444	16,824.05	5.63	17.31	88.10	−0.407	Cytoplasmic
15	ch04	1,496,628	1,497,874	reverse	147	444	16,770.01	5.72	35.06	87.41	−0.365	Cytoplasmic
16	ch04	1,505,284	1,506,417	forward	134	405	15,762.00	5.64	56.47	92.84	−0.653	Cytoplasmic
17	ch04	49,064,622	49,065,920	reverse	147	444	17,152.77	5.90	36.51	91.43	−0.351	Cytoplasmic
18	ch05	58,320,136	58,321,632	reverse	150	453	17,596.99	5.22	50.34	96.00	−0.467	Cytoplasmic
19	ch05	58,346,771	58,347,729	reverse	150	453	17,698.46	5.47	51.29	86.93	−0.456	Cytoplasmic
20	ch05	58,352,298	58,352,492	reverse	64	195	7600.66	5.18	43.07	83.75	−0.502	Cytoplasmic
21	ch05	58,352,813	58,353,016	reverse	67	204	7959.23	7.06	33.19	92.99	−0.058	Cytoplasmic
22	ch05	58,415,437	58,415,727	reverse	96	291	11,409.16	5.04	54.80	89.38	−0.431	Cytoplasmic
23	ch05	58,420,502	58,420,723	reverse	73	222	8461.61	7.93	30.44	77.26	−0.458	Cytoplasmic
24	ch05	64,304,632	64,306,501	reverse	162	489	18,181.90	4.78	37.17	89.63	−0.188	Cytoplasmic
25	ch06	1,757,744	1,758,229	forward	161	486	17,717.56	5.23	33.57	106.40	0.131	Cytoplasmic
26	ch07	288,360	289,609	reverse	149	450	16,770.27	4.73	25.47	99.33	−0.013	Cytoplasmic
27	ch07	291,108	292,521	forward	158	477	17,981.36	5.23	37.64	87.47	−0.285	Cytoplasmic
28	ch07	3,683,904	3,685,158	forward	159	480	18,422.81	5.31	32.72	75.91	−0.613	Cytoplasmic
29	ch08	26,737,023	26,738,299	reverse	148	447	17,074.62	6.50	16.91	88.92	−0.42	Cytoplasmic
30	ch09	296,718	297,754	reverse	148	453	17,074.62	6.50	16.91	88.92	−0.42	Cytoplasmic
31	ch09	301,066	301,598	reverse	136	411	15,514.72	5.09	22.29	95.96	−0.171	Cytoplasmic
32	ch09	304,321	305,404	reverse	150	453	17,083.46	4.88	28.19	96.07	−0.289	Cytoplasmic
33	ch09	339,412	340,474	forward	151	456	17,425.73	5.10	40.68	94.77	−0.438	Cytoplasmic
34	ch09	6,149,317	6,151,058	reverse	150	453	17,100.71	5.57	25.66	92.87	−0.223	Cytoplasmic
35	ch09	6,153,756	6,154,061	reverse	101	306	11,770.43	8.44	34.15	73.27	−0.716	Cytoplasmic
36	ch09	6,176,738	6,177,747	reverse	152	459	17,414.09	5.47	24.29	88.29	−0.266	Cytoplasmic
37	ch09	6,195,377	6,197,648	reverse	74	225	8310.73	8.80	24.74	75.00	−0.176	Cytoplasmic
38	ch09	6,221,354	6,222,253	reverse	152	459	17,423.11	5.62	31.78	90.26	−0.248	Cytoplasmic
39	ch09	70,346,085	70,347,506	reverse	155	468	17,235.60	5.79	22.44	84.84	−0.497	Cytoplasmic
40	ch09	70,350,133	70,352,154	reverse	160	483	17,368.77	5.44	30.62	94.38	−0.128	Cytoplasmic
41	ch09	70,355,642	70,357,036	reverse	160	483	17,914.20	5.34	32.81	88.25	−0.426	Cytoplasmic
42	ch09	70,360,338	70,361,437	forward	155	468	17,352.86	5.67	27.16	86.06	−0.389	Cytoplasmic
43	ch10	2,466,093	2,467,665	reverse	149	450	17,272.9	8.74	25.91	72.01	−0.606	Cytoplasmic
44	ch10	43,660,724	43,661,266	reverse	146	441	16,685.34	6.37	13.79	85.48	−0.305	Cytoplasmic
45	ch12	65,664,355	65,665,560	forward	158	477	18,196.91	5.03	38.7	99.94	−0.218	Cytoplasmic

**Table 2 ijms-23-01502-t002:** The gene ontology (GO) term distribution of PR-10 proteins of tomato.

Molecular Functions	Biological Process	Cellular Component
Protein binding	Immune system process	Cytosol
Lipid binding 44.9%	Response to stress 41.4%	Nucleus
Lyase activity	Reproduction	Organelle
Ion binding	Anatomical structure development	Vacuole
Nuclease activity	Cellular nitrogen compound metabolic process	Cytoplasmic-membrane-bounded vesicle
Hydrolase activity	Signal transduction	Cytoplasm 17%
N/A 28.9%		Cell
		Intercellular
		N/A

**Table 3 ijms-23-01502-t003:** Disease incidence and index level of TSWV in the CK and treatment (TR) groups of M82 and H8 tomato cultivars (highly resistance (0 < disease index ≤ 5); medium resistance (5 < disease index ≤ 20); resistance (20 < disease index ≤ 40); highly sensitive (60 < disease index); sensitive (40 < disease index ≤ 60)).

Post Inoculation of TSWV	CK Group		TR Group	
	cv. M82 (Sensitive)	cv. H8 (Resistant)	cv. M82 (Sensitive)	cv. H8 (Resistant)
14 days	10.0% ^c^ 5.3 ^d^	3.33% ^c^ 0.0 ^e^	16.67% ^c^ 12.5 ^d^	10.0% ^c^ 3.0 ^f^
21 days	26.67% ^b^ 46.66 ^b^	10.0% ^c^ 5.4 ^d^	36.67% ^b^ 52.3 ^b^	10.0% ^c^ 8.3 ^e^
28 days	40.0% ^a^ 58.88 ^a^	26.66% ^b^ 14.0 ^c^	100% ^a^ 88.88 ^a^	33.33% ^c^ 18.33 ^c^

**Note:** Same letters represent no significant difference, while different letters represent significant difference between CK and TR groups of M82 and H8 plants of TSWV disease incidence and disease index level, after TSWV inoculation at 14, 21 and 28 days. TSWV disease incidence expressed in percentages (%) and disease index express HR, MR, S, and HS respectively. Means ± SD at *p* < 0.05 using Tukey’s test with three biological replicates.

**Table 4 ijms-23-01502-t004:** Plant disease index level parameters.

Grade Series	TSWV Symptoms
0	Asymptomatic
1	Leaves are less obvious bronze (black) small spots
2	Bronze (black) necrosis spots spread in small area of leaves
3	Plant leaves on the film even into pieces, nearly 2/3 leaf area dry necrosis
4	Whole plant necrosis

**Table 5 ijms-23-01502-t005:** The correlation coefficient between PR-10 protein and *Sw-5b* at different time points after TSWV inoculation.

Index	CK	3 h	6 h	12 h	24 h	48 h	72 h	96 h	7 D	14 D	21 D	28 D
TR	1.000											
3 h	0.495	1.000										
6 h	0.550	0.926 **	1.000									
12 h	0.460	0.952 **	0.960 **	1.000								
24 h	0.123	0.692 **	0.489	0.606 **	1.000							
48 h	0.516	0.850 **	0.959 **	0.923 **	0.341	1.000						
72 h	0.154	0.764 **	0.590 **	0.713 **	0.975 **	0.456	1.000					
96 h	0.363	0.869 **	0.853 **	0.902 **	0.785 **	0.820 **	0.841 **	1.000				
7 D	0.314	0.839 **	0.847 **	0.820 **	0.527	0.822 **	0.602 **	0.777 **	1.000			
14 D	0.322	0.821 **	0.799 **	0.752 **	0.480	0.717 **	0.565	0.636 **	0.926 **	1.000		
21 D	0.200	0.616 **	0.651 **	0.659 **	0.479	0.720 **	0.545	0.782 **	0.799 **	0.672 *	1.000	
28 D	0.323	0.581 **	0.718 **	0.675 **	0.374	0.772 **	0.447	0.777 **	0.715 **	0.587 *	0.923 **	1.000

* Correlation is significant at the 0.05 level; ** correlation is significant at the 0.01 level.

## Data Availability

The data presented in this study are available on request from the corresponding author.

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
