# Peer review of "Genome-Wide Identification and Functions against Tomato Spotted Wilt Tospovirus of PR-10 in Solanum lycopersicum"

_ijms, 2022, doi:10.3390/ijms23031502_

Round 1

Reviewer 1 Report

Islam et al. have presented results on Genome Wide Identification and Functions against Tomato spotted wilt tospovirus of PR-10 in Solanum lycopersicum. The manuscript needs a substantial revision to be published in a journal like IJMS. I have mentioned a few not all the deficiencies of the manuscript. Firstly, the English language needs to be corrected by a native English speaker (Line 16, Line 20-21 and so on). Abstract starts with TSWV. Its full form should be provided when used for the first time, both in abstract and introduction. Abstract should be rewritten clearly mentioning the aim of the manuscript.  The experiment that has been done to obtain results should be mentioned in short in the abstract. Otherwise, the results mentioned in the abstract are not giving a proper interpretation. The authors should add few lines about the benefit of the manuscript/work and its novelty in the abstract. They can add that how the presented data will be beneficial for further research work in this direction. The novelty of this manuscript should be highlighted in both the abstract and introduction. Different sections written in the introduction do not seem to be connected to each other. It should be rearranged to make the aim of study clear to the readers. The grinding of diseased leaf sample in 2.2.7 should definitely come after the treatment and sampling that is 2.2.10. In results, Figure 3 is not clear. Its labels should be readable. Chromosome mapping part is missing from the method. The clusters obtained in Figure 7 have not been explained anywhere. ‘a’ of Figure 8 seems confusing. The plants did not seem inoculated. However, it is written after inoculation. Authors should discuss the results and how they can be interpreted from the perspective of previous studies and of the working hypotheses. The findings and their implications should be discussed in the broadest context possible. I believe that manuscript should be properly revised before accepting for publication.

Author Response

Response to Reviewer 1 Comments

Point 1: Islam et al. have presented results on Genome Wide Identification and Functions against Tomato spotted wilt tospovirus of PR-10 in Solanum lycopersicum. The manuscript needs a substantial revision to be published in a journal like IJMS. I have mentioned a few not all the deficiencies of the manuscript. Firstly, the English language needs to be corrected by a native English speaker (Line 16, Line 20-21 and so on). 

Response 1: Rewrite of abstract have been done the previous line 16, 20-21 and so on and also English editing the whole paper.  

Point 2: Abstract starts with TSWV. Its full form should be provided when used for the first time, both in abstract and introduction. Abstract should be rewritten clearly mentioning the aim of the manuscript.  The experiment that has been done to obtain results should be mentioned in short in the abstract. 

Response 2: TSWV write in the abbreviate form in abstract and introduction. Abstract already rewritten clearly mentioning the aim of the abstract and experiment results are mentioned in the abstract.

Point 3: Otherwise, the results mentioned in the abstract are not giving a proper interpretation. The authors should add few lines about the benefit of the manuscript/work and its novelty in the abstract. They can add that how the presented data will be beneficial for further research work in this direction. The novelty of this manuscript should be highlighted in both the abstract and introduction. 

Response 3:  Last paragraph of the abstract “our study provides comprehensive information on the members of PR10 gene families especially PR10 (Solyc09g090980) along with Sw-5b that will help in elucidating their exact function in tomatoes”. This is the benefit of the manuscript and its novelty in the abstract.

Point 4: Different sections written in the introduction do not seem to be connected to each other. It should be rearranged to make the aim of study clear to the readers. 

Response 4: Rewritten the abstract

Point 5: The grinding of diseased leaf sample in 2.2.7 should definitely come after the treatment and sampling that is 2.2.10. 

Response 5:  2.2.7 is TSWV stock solution preparation

Point 6: In results, Figure 3 is not clear. Its labels should be readable. Chromosome mapping part is missing from the method. The clusters obtained in Figure 7 have not been explained anywhere. ‘a’ of Figure 8 seems confusing. The plants did not seem inoculated. However, it is written after inoculation. Authors should discuss the results and how they can be interpreted from the perspective of previous studies and of the working hypotheses. The findings and their implications should be discussed in the broadest context possible.

Response 6: figure 3 is clear and add chromosome mapping part in the methods. Figure 7 explian in the results portion and figure 8 add original new picture that would be not confusing. Rewrite the results and discussion portion that would be helpful for the readers.

Reviewer 2 Report

The manuscript “Genome Wide Identification and Functions against Tomato spotted wilt tospovirus of PR-10 in Solanum lycopersicum” studies the PR-10 genes in Solanum lycopersicum. The authors identified the putative PR-10 genes in tomato taking the comparative genomics approach and genome – wide identification. They identified 45 candidate genes, their exon-intron structures, protein chemical and physiclal properties, localization and annotation. In addition to studying their transcriptional profile in tomato under control and infected conditions. Generally, the manuscript is well designed and readable, but the figures are not clear. 

I will recommend doing minor edits (comments attached in the manuscript files) before publishing.

Author Response

Response to Reviewer 2 Comments

Point 1:  TSWV in the abstract portion

Response 1:  Start write in the abbreviate form

Point 2:  “Forty-five PR10 gene candidates of amino acid sequences” in the materials section  

Response 2:   write Forty-five amino acid sequences of PR10 encoding genes

Point 3:  e-value threshold of 1e-10

Response 3:  write e-value threshold of 1e-10

Point 4:  please remove the number of transcript from all gene IDs in this column

Response 4:  Already remove the gene IDs from the table 2.

Point 5:  gene IDs and size of the strand in this figure are not clear, please redo this figure with clear text and white background

Response 5: add new figure with gene IDs and size of the strand with clear text, white background

Point 6:  text is not clear please increase the resolution of this figure. also, the white background is preferable

Response 6 : add new figure with gene IDs with white background

Point 7:  the text of IDs are not clear, please increase the distance between rows in the heatmap and remove the number of the transcripts of the gene IDs. please indicate what is shown in the write panel of the figure.

Response 7: Increase this picture quality of figure 7 and increase the distance between rows in the heatmap. Here we show that PR10 gene (Solyc09g090980) highly express into different parts of the tomato and remove the right panel of the figure.

Point 8:  please add information about the sttatistical analyses and please do the letters of significance as superscript

Response 8: Add information the statistical analysis and do the letters of significance as superscript in table 4.

Round 2

Reviewer 1 Report

As the authors have made the suggested modifications, the manuscript can be accepted in its present form.